# Translocation of Drought-Responsive Proteins from the Chloroplasts

**DOI:** 10.3390/cells9010259

**Published:** 2020-01-20

**Authors:** Ping Li, Haoju Liu, Hong Yang, Xiaojun Pu, Chuanhong Li, Heqiang Huo, Zhaohui Chu, Yuxiao Chang, Yongjun Lin, Li Liu

**Affiliations:** 1National Key Laboratory of Crop Genetic Improvement and National Centre of Plant Gene Research, Huazhong Agricultural University, Wuhan 430070, China; liping@mail.kib.ac.cn (P.L.); liuhaoju@mail.jxau.edu.cn (H.L.);lchh5@webmail.hzau.edu.cn (C.L.); 2Key Laboratory for Economic Plants and Biotechnology, Germplasm Bank of Wild Species, Kunming Institute of Botany, Chinese Academy of Sciences, Yunnan Key Laboratory for Wild Plant Resources, Kunming 650201, China; yanghong@mail.kib.ac.cn (H.Y.); puxiaojun@mail.kib.ac.cn (X.P.); 3Mid-Florida Research and Education Center, Department of Environmental Horticulture, University of Florida, Miami, FL 32703, USA; hhuo@ufl.edu; 4State Key Laboratory of Crop Biology, Shandong Provincial Key Laboratory of Agricultural Microbiology, Shandong Agricultural University, Taian 271018, China; zchu@sdau.edu.cn; 5Agricultural Genomics Institute at Shenzhen, Chinese Academy of Agricultural Sciences, Shenzhen 518120, China; yuxiao.chang@outlook.com; 6State Key Laboratory of Biocatalysis and Enzyme Engineering, Hubei Collaborative Innovation Center for Green Transformation of Bio-Resources, Hubei Key Laboratory of Industrial Biotechnology, School of Life Sciences, Hubei University, Wuhan 430070, China

**Keywords:** chloroplast, ITRAQ, drought, proteomics, retrograde signal

## Abstract

Some chloroplast proteins are known to serve as messengers to transmit retrograde signals from chloroplasts to the nuclei in response to environmental stresses. However, whether particular chloroplast proteins respond to drought stress and serve as messengers for retrograde signal transduction are unclear. Here, we used isobaric tags for relative and absolute quantitation (iTRAQ) to monitor the proteomic changes in tobacco (*Nicotiana benthamiana*) treated with drought stress/re-watering. We identified 3936 and 1087 differentially accumulated total leaf and chloroplast proteins, respectively, which were grouped into 16 categories. Among these, one particular category of proteins, that includes carbonic anhydrase 1 (CA1), exhibited a great decline in chloroplasts, but a remarkable increase in leaves under drought stress. The subcellular localizations of CA1 proteins from moss (*Physcomitrella patens*), *Arabidopsis thaliana* and rice (*Oryza sativa*) in *P. patens* protoplasts consistently showed that CA1 proteins gradually diminished within chloroplasts but increasingly accumulated in the cytosol under osmotic stress treatment, suggesting that they could be translocated from chloroplasts to the cytosol and act as a signal messenger from the chloroplast. Our results thus highlight the potential importance of chloroplast proteins in retrograde signaling pathways and provide a set of candidate proteins for further research.

## 1. Introduction

Drought stress, which adversely affects plant growth and causes substantial losses in crop production [1], has been exacerbated by climate change. Plants have developed various mechanisms to respond to stress, involving changes in gene expression, metabolism, and physiology [2,3,4]. In addition, plants have complex networks for sensing and signaling environmental stress using different hierarchies of sensors including those in organelles such as chloroplasts [5].

In one stress-sensing/signaling pathway, perturbation of cellular homeostasis due to environmental stress is reported to the nucleus from organelles by so-called retrograde signals [5]. These organelle-derived signals regulate transcriptional activities in the nuclei through communication pathways whose mechanism is unclear [6]. Given that chloroplasts are major sites for the biosynthesis of amino acids, secondary metabolites, and phytohormones [7,8], it poses the possibility that some of these molecules may serve as signals for communication from the plastids to the nuclei; for example, 2-c-methyl-d-erythritol-2,4-cyclo-diphosphate (MEcPP) and phosphor-nucleotide 3-phosphoadenosine 5-phosphate (PAP) were reported to serve as retrograde signal molecules [9,10]. In addition, the main tetrapyrroles in chloroplasts, including chlorophyll, heme, siroheme, and phytochromobilin, could function as intermediates in the communication between chloroplasts and the nuclei [10,11,12,13,14].

Like the identified signaling molecules, protein metabolism is also responsive to environmental stresses [15]. For example, defects in the chloroplast-targeted RH3 DEAD-box protein in *Arabidopsis* inhibit the maintenance of abscisic acid levels under environmental stresses [16,17]. In addition, mutations of chloroplast proteins HCF106 and THF1 lead to higher ROS accumulation in guard cells, increased stomatal closure and stronger drought resistance in *Arabidopsis thaliana* [18]. The chloroplast-localized carbonic anhydrase 1 (CA1) also functions in the early CO_2_ signaling pathway and influences water use efficiency in *Arabidopsis* leaves [19]. Recently, a newly identified chloroplast protein, glycosyltransferase QUA1, has been demonstrated to play a crucial role in chloroplast-dependent calcium signaling under salt and drought stresses [20]. These findings raise the question of how these signals are distinguished by the nucleus from the signals generated from other organelles to initiate signaling cascades that trigger source-specific nuclear responses. Wang et al. (2014) [21] observed that some stress-induced proteins were transferred out of chloroplasts to the cytosol through the process of chloroplast vesiculation, indicating that chloroplast-derived proteins may possibly act as signal messengers to trigger cascade reactions. However, the type of proteins that serve as signal messengers under drought stress needs further investigation.

Isobaric tags for relative and absolute quantitation (iTRAQ) is an isobaric labeling method coupled with tandem mass spectrometry for proteome analysis [22]. This method has been used for studying plant growth and stress tolerance by comparing protein abundance in different organs. The proteomics studies in maize (*Zea mays*), cotton (*Gossypium* sp.), poplar (*Populus cathayana*), and soybean (*Glycine max*) have established that iTRAQ is effective and reliable method for proteome analysis under stress [22,23,24] [25]. Lei et al. (2018) [26] used iTRAQ to determine changes in chloroplast proteins of tobacco leaves treated with Cucumber Mosaic Virus, and they found that iTRAQ is feasible for investigating the global changes in protein accumulation in chloroplasts as well as in whole cells.

Understanding the mechanisms behind how plants sense and acclimate to drought stress is fundamental for improving stress tolerance and for sustainable agricultural production. Therefore, a better understanding of how retrograde signals originating in the chloroplast are transferred to the nucleus will be crucial for elucidating intracellular coordination to regulate gene expression in both the nucleus and chloroplasts for physiological responses. In this study, to further explore whether chloroplast proteins could serve as potential retrograde signal messengers in drought stress, we used the iTRAQ method to examine proteins with differential abundance in the leaf and isolated chloroplasts during drought stress and recovery in *Nicotiana benthamiana*. We identified 3936 and 1087 proteins with altered abundance in the leaf (total proteins, TPs) and chloroplasts (chloroplast proteins, CPs), respectively, under drought and re-watering treatments. Of particular interest was one category of proteins that declined in abundance in the chloroplasts under drought stress, but significantly increased in the total leaf proteins (cytosol + chloroplasts), which we called PSAPs (proteins with special accumulation pattern). These proteins represent potential retrograde signal messengers since they were induced by drought and transferred into the cytosol or nucleus for potential signal transduction. To further explore their function, we investigated the subcellular localization of a moss (*Physcomitrella patens*), a *A. thaliana* and a rice (*Oryza sativa*) homolog of one of these proteins, carbonic anhydrase 1 (CA1), in *P. patens* protoplasts under osmotic stress. Our results provide evidences supporting the possible function of proteins as messengers in retrograde signaling pathways.

## 2. Materials and Methods

### 2.1. Plant Growth Conditions and Treatments

*Nicotiana benthamiana* seedlings (a gift from Dr. Yongjun Lin in Huazhong Agricultural University) were grown in potting soil with 3L plastic pots in a greenhouse with a12 h light/12 h dark cycle, and a light intensity of 180 μmol m^−2^ s^−1^ at 22 °C. When seedlings reached 40-day-old, irrigation was discontinued for 8 days (D0–D8) to induce drought stress, as described by [27]. During the drought treatment, the water content of individual plants was monitored daily by subtracting the weight of the pots and soil from the weight of the seedlings; multiple plants and pots were used to obtain mean water contents. After the 8 day drought treatment, plants were re-watered for two days (R1: re-watered for 8 h; R2: re-watered for 24 h; R3: re-watered for 48 h). Relative water content (RWC) was calculated using the formula [1], TW represents the total weight of the non-treated plants, the soil, and the pots; FW represents the total weight of the treated plants, the soil, and the pots), and DW represents the total weight of the soil and pots [28]. The net photosynthetic rate, intercellular CO_2_ concentration, transpiration rate, and stomatal conductance of different treatment tobacco were monitored Li-6400XT (Li-Cor Biosciences, Lincoln. NE. USA) [29]. The total chlorophyll was extracted from the treatment tobacco leaves with extracted buffer (50% acetone and 50% ethanol) and measured with spectrophotometer (Yiheng, Shanghai, China) [30].
RWC (%) = ((FW-DW)/(TW-DW)) × 100(1)

### 2.2. Quantification of Phytohormones

Phytohormones (ABA) were extracted from 0.1–0.3 g frozen *N. benthamiana* leaves with the method developed by Cai et al.(2015) [31]. ABA (0.6 ng) were added to the sample extraction buffer as a standard. Quantifications of phytohormone were conducted using high-performance liquid chromatograph mass spectrometer (LCMS-8040, SHIMADZU, Japan) according to the protocol described by Lee et al. (2015) [32].

### 2.3. Preparation of Protein Samples

Total protein and chloroplast proteins were isolated from plants with different water contents after the drought and re-watering treatments. Leaf samples (10 g) were pooled from multiple plants, flash-frozen in liquid nitrogen and stored under −80 °C until use for total protein isolation. For chloroplast protein extraction, 15 g fresh leaves from plants with different treatments were pooled and chloroplast were isolated following the protocol developed by Wang et al. (2013) [33] with minor modifications. Briefly, leaf tissues were mixed in buffer I (0.05 M Hepes, 0.35 M sorbitol, 1 mM MgCl_2_), ground with a shredder machine (Waring 8011S, USA), and homogenized samples were filtered through four layers of medical gauze (Solarbio, Beijing, China) to isolate the chloroplast-containing solutions. Chloroplasts were then harvested by spinning at 1500× *g* for 5 min with the brake off and carefully resuspended in 2 ml buffer I; the resuspended chloroplasts were transferred onto the upper of a percoll gradient solution. Chloroplasts were further separated through the gradient by centrifugation at 1500× *g* for 20 min with the brake off, and the middle layer between 40% and 80% percoll was harvested for downstream analysis (Appendix A). Chloroplasts were washed 2 times with cold buffer II (0.1M K-Tricine, 0.66 M sorbitol, 6 mM MgCl_2_, pH 8.0) (Appendix A).

Proteins were extracted from leaves and chloroplasts using Arkadiusz’s method with minor modifications. In brief, proteins were isolated using buffer III (4% w/v SDS, 100 mM Tris-HCl, 1 mM DTT, pH 7.6). The protein concentration was determined with the commonly available bicinchoninic acid (BCA) assay as described by Fan et al. (2015) [34].

### 2.4. Western Blot Analysis

The proteins were separated by 10% SDS-PAGE, then transferred to PVDF membranes (GE, USA) using a wet transblot system- Criterion (Bio-Rad, USA). The membrane was incubated with antibodies against ribulose-1,5-bisphosphate carboxylase/oxygenase (Rubisco large subunit), alternative oxidase (AOX) (Agrisera, Sweden), Hyd (alpha/beta-Hydrolases superfamily protein, NbS00028999g0008.1), NPQ1 (NPQ1, NbS00013338g0011.1), Mog1 (Mog1/PsbP/DUF1795-like photosystem II reaction center PsbP family protein, NbS00055576g0008.1), PRPL11 (PRPL11, NbS00029944g0006.1) and L1p (Ribosomal protein L1p/L10e family, NbS00005939g0007.1) at 1:2000 dilutions at room temperature for 6 h. The membranes were stained with goat antirabbit IgG at 1:1000 dilution for 2 h at room temperature, after washing three times with TBST buffer (0.01M TBS, 0.1% v/v Tween-20, pH 7.6). The results were obtained using MicroChemi 4.2 Bio-imaging Systems (DNR, ISRAEL). The experiments were repeated three times.

### 2.5. ITRAQ Labeling and LC-MS/MS Assay

Protein samples were labeled with 8-derivatization reagents from the AB SCIEX kit (USA) [35]. Three biological replicates were applied to each sample. The sample preparation and digestion were performed following the method of Xie et al. (2016) [36]. LC-MS/MS analysis were conducted on the labeled samples with Q Exactive’s HCD (Higher Energy Collisionsl Dissociation) model (Thermo Scientific, USA). The raw data of Q Exactive were transformed with proteome discoverer 1.2 (Thermo Scientific) and submitted to MASCOT2.2 for data analysis. Credible qualitative results were screened using a false discovery rate (FDR) < 0.01 with MASCOT2.2. The screened peptides were used for BLAST searches with the function of MASCOT2.2 against the *N. benthamiana* protein database at NCBI. The intensity values of ion peaks were extracted and quantitative values were normalized with Xcalibur Proteome Discoverer 1.4 following the instruction manual (Thermo Scientific, USA).

### 2.6. Bioinformatic Analysis

To analyze the function of all target proteins, we conducted Gene Ontology (GO) annotation. Conserved motifs were searched against the European Bioinformatics Institute (EBI) database with InterProScan [37] and the significance of GO terms was determined with Fisher’s Exact Test. The query proteins were BLAST searched against Kyoto Encyclopedia of Genes and Genomes (KEGG), the homologs were cut off based on the Bi-Directional Hit Rate, then the orthologous candidates were clustered through KEGG by scoring with Probability and Heuistics, http://www.ebi.ac.uk/InterProScan. The significant metabolic and signaling pathways were determined with Fisher’s Exact Test for KEGG enrichment analysis [38]. The identified proteins were clustered based on their expression pattern using the Short Time-series Expression Miner (STEM) version 1.3.11. The phylogenetic relationships were performed using MEGA 6.06 with the Neighbor-Joining (NJ) method. The conserved motifs were identified using MEME [39]. Protein structures analysis were using I-TASSER software [40].

### 2.7. Plasmid Construction and Transformation

The genes of rice (*O. sativa*) and *P. patens* encoding homologs of the *N. benthamiana* protein NbS00037492g0013.1 identified in this study were retrieved through BLAST analysis against the respective databases at Phytozome 12 (https://phytozome.jgi.doe.gov). To construct the 35S::GFP gene expression vector, the full-length cDNAs of Pp3c24_9430 from *P. patens* and LOC_Os01g45274 from *O. sativa* were synthesized with the cDNA synthesis kit (Takara, cat: R047A) and ligated into the pCAMBIA2300-GFP binary vector [41]. The transient PUB:35S-GFP gene expression vector and the pGC1: CA-II-YFP overexpressing *Arabidopsis* were coming from Hu lab [42]. The plasmids were transformed into *P. patens* protoplasts with the PEG-mediated method developed by NIBB lab [43]. Protoplasts were treated with a 50 mM NaCl solution (Solarbio, China) for about 30 min to generate osmotic stress after transformation. The *Arabidopsis* lines were treated with 200 mM NaCl and 400 mM Mannitol for 3 h and longer time (4 d, 14 d).

## 3. Results

### 3.1. Physiological Changes in Nicotiana Benthamiana Plants under Drought Stress

To examine the physiological changes of *N. benthamiana* plants in response to drought stress, 40 day-old *N. benthamiana* plants were subjected to drought stress, and their relative water content (RWC), and phytohormone were determined. The RWC significantly decreased after more than 4 day drought treatment (*p* ≤ 0.01). After an 8 day drought treatment (denoted by D0–D8), the RWC had decreased to 70% (Figure 1a). After the 8 day drought treatment, plants were re-watered for two days (R1: re-watered for 8 h; R2: re-watered for 24 h; R3: re-watered for 48 h). There are no obvious leaf senescence and degradation of chlorophyll during the stress treatment according to the physiology parameters (Appendix A).

We also examined the effect of drought stress on the content of endogenous hormones. ABA is well known as a stress response signal; its level dramatically increased during the drought stress treatment at D5 and remained high even at R1 (Figure 1b). The levels of ABA decreased after two days re-watering (R3).

### 3.2. Differentially Accumulated Proteins (DAPs) in the Leaf and Chloroplasts under Drought Stress

To screen for drought-responsive proteins, we chose D0, D1, D5, R1, and R3 for further analysis according to the physiological results, and quantified the CPs and TPs after the drought treatments using the iTRAQ-based protein quantification method with three replicates for each sample as diagramed in Figure 2. Purity assessment of chloroplast proteins and total proteins was performed through Western blot with antibodies specific for Rubisco and AOX (Appendix A). The reliability of our data was verified using antibodies of some differentially accumulating proteins (Appendix A). After analyzing with liquid chromatography-tandem mass spectrometry (LC-MS/MS), we identified 5,230 TPs and 1,522 CPs. Among these proteins, 3936 TPs and 1087 CPs, quantified with at least two repeated test, showed significant changes in abundance between the drought and re-watering treatments with an abundance ratio ≥1.2 or ≤0.83 (Appendix A). The differentially accumulating proteins were obtained by comparing D0 and D1, D0 and D5, D0 and R1, D0 and R3, D1 and D5, D1 and R1, D1 and R3, D5 and R1, D5 and R3, and R1 and R3. Many of these differentially abundant proteins are related to ABA, which reflected their response to drought treatments and the endogenous ABA change (Appendix A). In addition, some proteins that reported to be involved in regulating drought stress resistance, were also accumulated under drought stress (Appendix A). We found that 39 ROS-associated proteins, 47 HSPs and 29 transcription factors also affected by drought stress (Appendix A).

The differentially abundant proteins were subjected to gene ontology (GO) annotation and KEGG enrichment. The most differentially abundant TPs mainly fell into functional categories for biosynthesis and processing of amino acids, ribosomes, and carbon metabolism, which accounted for 49.34% of the differentially abundant TPs. These findings indicated that drought stress triggered adaptive systems, and that some constitutive processes were preferentially sustained to provide substances for maintaining normal growth (Figure 3a). The second most abundant category of differentially abundant TPs were involved in processes of carbon fixation, oxidative phosphorylation, glycolysis/gluconeogenesis metabolism, photosynthesis, glyoxylate and dicarboxylate metabolism, and TCA cycles, which accounted for 28.75%. This pattern indicated that basic physiological metabolic pathways were induced by drought stress to maintain a normal energy supply (Figure 3).

In chloroplasts, most proteins involved in metabolism were significantly induced by drought stress, similar to the TPs in the leaves (Figure 3b). Porphyrin and chlorophyll metabolism, ATP synthesis, carbon fixation, and carotenoid biosynthesis were the most abundant functional groups in both TPs and CPs. However, the quantity and fold change were different, with most CPs showing larger changes than TPs, such as OHP2 (NbC25835994g0001.1), PSAF (NbS00000058g0018.1), and so on (Appendix A).

### 3.3. The Abundant Pathways Induced by Drought Stress in Leaves and Chloroplasts

In contrast to the upregulated proteins under drought stress, some proteins exhibited significant reduction under drought stress, such as phosphatases and kinases for glycolysis/gluconeogenesis metabolism.

As Figure 4a shows, proteins related to glycolysis/gluconeogenesis metabolism were significantly accumulated under drought stress. The rapid and significant induction of phosphoglucomutase (PGM), glucose-6-phosphate isomerase (GPI), fructose-1,6-bisphosphatase I (FBP), 6-phosphofructokinase (PFK), glyceraldehyde 3-phosphate dehydrogenase (GAPDH), phosphoglycerate kinase (PGK), enolase (ENO), and pyruvate kinase (PK) likely ensures the continued production of energy and provides signals for related metabolism. Glucose produced from carbon fixation is the precursor of pentose phosphate in glycolysis/gluconeogenesis metabolism. Glycolysis can produce β-fructose-6-phosphate (D-Fru-6p), which serves as the main precursor of histidine synthesis. Pyruvate generated through glycolysis can be transported to mitochondria for oxidative phosphorylation and the TCA cycle. Catalase (CAT) and superoxide dismutase (SOD) were also induced by drought, possibly functioning in ROS scavenging.

As Figure 4b shows, most enzymes for carbon fixation, photosynthesis, porphyrin, and chlorophyll metabolism, and carotenoid biosynthesis were also significantly induced by drought stress, especially in chloroplasts.

Of particular interest were proteins that decreased in chloroplasts but increased in leaf cytosol under drought stress, suggesting their potential as signaling messengers under drought stress. These proteins included phosphoglycerate kinase (PGK), plastid transketolase (E2.2.1.1), and sedoheptulose-bisphosphatase (E3.1.3.37). A similar pattern was also observed for most proteins related to porphyrin and chlorophyll metabolism, which rapidly decreased in chloroplasts immediately after drought stress, but were elevated in leaves. This result suggested that porphyrin and chlorophyll metabolism might also be critical for drought tolerance. This was not surprising because the products of porphyrin and chlorophyll metabolism act as the upstream precursors for synthesizing stress regulators like ABA, JA, and others. Additionally, porphyrin and chlorophyll metabolism directly affect the light-dependent reaction of photosynthesis, resulting in the efflux of ATP and energy (Figure 4b).

### 3.4. PSAPs Induced by Drought Stress in TPs and CPs

To further profile the differentially abundant proteins in leaves and chloroplasts induced by the drought treatment, the proteins were classified based on their accumulation patterns. A total of 3936 TPs and 1087 CPs were grouped into 16 profiles, respectively. The TPs with differential accumulation patterns were divided into two major groups (Figure 5). TPs in profiles 4, 5, 10, and 16 were increased in leaf cytosol under drought stress; however, the total amounts of proteins in profiles 10, 5 and 4 were reduced at D5, R1, and R3, respectively. Therefore, we termed this group of TPs “up-down TPs”. TPs in profiles 3, 9, 14, and 15 were first decreased at earlier stages of drought stress, but proteins in profiles 14, 9, and 3 increased at D5, R1, and R3, respectively. We termed this group of proteins as “down-up TPs”. Some of the accumulation patterns were complex. For example, proteins in profiles 7, 12, and 13 were induced by drought, followed by a reduction at R1, D5, and R1, but were elevated again at R3, respectively. Similar results were also observed in profiles 6, 11, and 8 with an initial reduction at the earlier stages of drought stress treatment, followed by an increase at D5, D5, and R1, then a second reduction at R3, R1, and R3, respectively. Other protein accumulation patterns were even more complicated. The accumulation patterns of proteins in profiles 2 and 1 had a reverse trend in each test. The proteins in profile 2 increased at D1, decreased at D5, increased at R1 and decreased at R3, while the dynamic pattern of proteins in profile 1 was the opposite. We found that 881 TPs were strongly induced by drought stress, and 772 of them were down-up TPs (Figure 5a). This dynamic of TPs is consistent with the first treatment of drought followed by a re-watering treatment.

To explore the source of these down-up TPs, we compared their abundance between TPs and CPs. In total, 231 proteins that decreased in chloroplasts during the end of drought treatment or the beginning of the re-watering treatment, but increased in leaf cells at the same time were selected for further analysis (Figure 5b, Appendix A). The increase of these proteins in TPs may result from the translocation of chloroplast proteins to the cytosol of leaves, and not by de novo protein synthesis in the cytosol under drought stress.

GO and KEGG analysis of the proteins that decreased in chloroplasts but increased in leaves (PSAPs) revealed that these proteins are mainly involved in pathways of photosynthesis, carbon metabolism, protein export and porphyrin and chlorophyll metabolism (Figure 5b). For example, glutamine synthetase (GS2, NbS00017344g0018.1) was induced by drought and re-watering treatment, decreased to 0.80 at R1 in chloroplasts, possibly due to relocation (Figure 5c, Appendix A). Malate dehydrogenase (MDH, NbS0004490g0002.1) increased from 1.38 (D0) to 1.40 (D1) and sharply decreased to 0.81 (R1) in chloroplasts, but increased to 1.06 (R3) in leaves. Two homologs of fructose-bisphosphate aldolase 2 had a similar alteration pattern: NbS00014581g0001.1 decreased from 1.41 (D1) to 1.07 (R1) in chloroplasts, but increased from 0.84 (D1) to 1.03 (R1) in leaves; NbS00028064g0012.1 decreased from 1.09 (D1) to 0.91 (R1) in chloroplasts, but increased from 0.84 (D0) to 1.11 (R1) in leaves. The same pattern was observed for Glutamate-1-semialdehyde 2, 1-aminomutase (GSA, NbS00057624g0013.1). One carbonic anhydrase (NbS00037492g0013.1) was significantly induced by drought (2.14 fold), then decreased at R1 (recovery) in chloroplasts, but this pattern was reversed in leaves. Notably, carbonic anhydrase functions in carbon metabolism, both inside and outside of the chloroplast (Figure 5b,c and Appendix A).

### 3.5. PSAPs Relocated under Drought Stress

To test the hypothesis that CPs may be relocated to the cytosol of leaves under drought stress, we used carbonic anhydrase 1 (CA1, NbS00037492g0013.1) to perform transient assays in *P. patens* protoplasts. Unfortunately, we were not able to obtain complete sequences of NbS00037492g0013.1 from the *N. benthamiana* genome; therefore, we used homologous proteins from rice (*O. sativa*), moss (*P. patens*) and *A. thaliana*. Six and two homologous proteins were found from moss (PpCA1) and rice (OsCA1) using the signature peptide of NbCA1, respectively (Table 1). Six and four homologous proteins were found from *A. thaliana* and *Solanum lycopersicum* (Figure 6a). Sequences analysis showed the carbonic anhydrase domain was conserved in the proteins, and the proteins of *S. lycopersicum* and *A. thaliana* had more similarity with NbS00037492g0013.1.

Protein sequence analysis with I-TASSER software showed that the conserved crystalized protein structures (1ddzA, 1ekjG, 401KA, 3ucnA, 5swcA, 5cxkA, 4rxyA, and 2w3qA) and ligand-binding sites (ACT-2w3Na and BCT-5cxkH) were shared between *N. benthamiana* NbS00037492g0013.1 and its homologs from *Arabidopsis*, moss and rice (AT3G01500.3, Pp3c24_9430, and LOC_Os01g45274) (Figure 6b). This suggested that the proteins were evolutionarily conserved and may share similar functions. To test the relocation of signals, we used homologous proteins of NbCA1 to perform the transient and stable assays.

The *Arabidopsis* β-CA1 has been identified in chloroplasts and in the vicinity of the plasma membrane [19]. Partly consistent with the previous study, after transformation in *P. patens* protoplasts, OsCA1-GFP (green fluorescent protein), PpCA1-GFP and AtCA1-GFP were localized in the chloroplasts (Figure 6c, Appendix A). However, under abiotic stress conferred by either 50 mM NaCl or 10 mM PEG 6000, green fluorescence signals from OsCA1-GFP and PpCA1-GFP first aggregated in the vicinity of the chloroplast membrane, and then appeared in the cell cytosol at a later stage (Figure 6c, Appendix A). The detached leaves of overexpressing *Arabidopsis* lines (pGC1: CA-II-YFP) were further treated with 400 mM Mannitol or 200 mM NaCl for 3 h, some YFP signals linked with CA1 (*Arabidopsis*) concentrated on some points of chloroplasts, but some YFP signals went through chloroplasts, and entered the cytoplasm, especially after the 14 day Mannitol treatment and four day NaCl treatment (Figure 6d, Appendix A). These results demonstrated that CA1 translocated outside the chloroplast under drought stress, and that the mechanism underlying this stress-induced protein translocation is conserved between moss, rice, and *Arabidopsis*.

## 4. Discussion

We identified sixteen differentially abundant protein profiles in response to drought and water-recovery treatments using iTRAQ and LC-MS/MS methods (Figure 2, Figure 5a). In total, 3936 proteins from leaves and 1087 proteins from isolated chloroplasts were significantly changed during treatments (Appendix A). A recent study of drought-responsive proteins in *N. benthamiana* reported 466 differentially abundant proteins, around 10-fold less than we discovered [36]. Due to the large number of differentially abundant proteins in our study, we drew a comprehensive map of the cellular activity response to drought and recovery. Most proteins involved in carotenoid biosynthesis, porphyrin and chlorophyll metabolism, carbon fixation, and calcium signaling were responsive early to drought stress. Most proteins involved in photosynthesis, pentose phosphate, and glycolysis/gluconeogenesis were responsive in later stages of drought stress. Proteins involved in the TCA cycle, oxidative phosphorylation, and peroxisome metabolism were responsive after recovery (Figure 3, Figure 4). During the drought-response stage, glucose and other products of carbon fixation likely connect the cellular activity of glycolysis and pentose phosphate metabolism for energy supply. D-Fru-6p and pyruvate produced in glycolysis connected the delayed responses to drought stress, such as the TCA cycle and oxidative phosphorylation. Proteins involved in the TCA cycle, oxidative phosphorylation, and peroxisome metabolism that were functional in the recovery phase, might potentially serve for scavenging oxygen radicals and maintaining physiological status.

As depicted in Figure 3, our KEGG pathway analyses identified several similar targets responsive to drought as those reported in tobacco and other plants [36,44,45]. However, we identified many novel targets, such as carbon fixation and metabolism, oxidative phosphorylation, pyruvate metabolism, TCA cycle, ribosome, metabolism of xenobiotics by cytochrome P450, and calcium signaling pathways. In particular, the differentially abundant CPs identified in our study provide a new, large pool of targets for studying the role of chloroplasts in responses to stress.

Approximately 80% of the detected proteins exhibited similar profiles between the TPs and CPs. However, 231 proteins showed quite different patterns when comparing their abundance in TPs and CPs (Appendix A). Some examples include fructose-bisphosphate aldolase, amino methyl transferase, and other important conserved proteins involved in carbon metabolism or photosynthesis. Among these, a large number of important enzymes corresponding to carbon metabolism have been identified in leaf senescence processes as well [46]. This is not surprising because transcriptome and protein networks connect or overlap during drought and senescence responses [47]. As the most basic process in living organisms, carbon metabolism, including carbon fixation and utilization, play key roles for ensuring a supply of energy during various stress responses [48,49]. It is reasonable to assume that proteins that participate in abiotic stress responses might function in activating leaf senescence. This study is in accordance with previous reports that leaf senescence and stress responses share common pathways involving cell death.

As centers for energy production and metabolic hubs, chloroplasts also communicate with the nucleus and mitochondria for maintaining normal biogenesis and cellular activity. Increasing evidences show that chloroplasts play important roles in the regulation of gene expression in response to abiotic stresses [50]. In recent years, several retrograde signals and signaling pathways were found to be related to stress resistance, such as the SAL1-PAP signaling pathway, and the production of MEcPP, photosynthesis-dependent hydrogen peroxide (H_2_O_2_), and carotenoid oxidation products [12,51,52]. A heterogeneously expressed antimicrobial peptide can be released from the intact chloroplast in responses to biotic and abiotic stresses, which had been visualized with confocal microscopy [53]. However, the mechanisms underlying the translocation of these chloroplast and endogenous macromolecules under stressful conditions are unknown. Our present proteomic analysis provided a large set of proteins to understand protein-mediated communication at the cellular level for dissecting these complicated intracellular signaling networks.

Considering the conserved function of chloroplast proteins among species, and incomplete *N. benthamiana* information, we chose rice, moss and *Arabidopsis* homologs of CA1 to explore the hypothesis of protein relocation during stress (Table 1). In agreement with the proteomic results, the rice, moss and *Arabidopsis* CA1 homologs moved out of the chloroplast after sodium and osmotic stress treatments of more than 30 min (Figure 6).

CAs function in accelerating the interconversion between CO_2_ and HCO_3_-, an activity essential for all organisms [49,54]. In *Arabidopsis*, βCAs function as upstream regulators in the CO_2_-induced stomatal signal transduction cascade based on impaired CO_2_ responses in the *ca1 ca4* double-mutant leaves. The *ca1 ca4* plants show impaired CO_2_ regulation of stomatal movements and increased stomatal density, while βCA1-overexpressing plants exhibit enhanced water use efficiency and result in the formation of extra tapetal cells [19,55,56]. In rice, βCA1 was demonstrated to be involved in carbon assimilation and the CO_2_-mediated stomatal pore response. Unlike in *Arabidopsis* and maize, knocking out *Os*β*CA1* decreased photosynthetic capacity [57,58]. βCA1 can bind to salicylic acid (SA), and probably initiate a signaling cascade to activate defense response genes [59,60]. Chloroplast-localized βCA1 plays important functions in carbon delivery for cellular activity, and also in diverse developmental and stress responses. Therefore, the question arises if the subcellular relocation of βCA1 relates to the regulation network via adjusting cellular stress response proteins or metabolite levels. Additional research will be needed to investigate the possible roles of βCA1 in response to biotic and abiotic stresses.

Unlike in vascular plants, there are no stomata or guard cells in moss leafy gametophores. Our results of moss βCA1 relocalization demonstrated that this chloroplast-localized CA shares a common function in response to stresses as it does in vascular plants, although it is not necessary for the control of stomatal movement. It will be interesting to investigate the regulated mechanisms of βCA1 in the course of evolution.

The present study presented a novel discovery that a large amount of chloroplast proteins may relocate to the cytosol in response to drought stress. Yet, it is not clear how the proteins moved out of the chloroplasts and what their function is in the cytosol. Therefore, further research is needed to confirm the function of these translocating proteins and discover their intracellular signaling networks.

## 5. Conclusions

We reported a novel discovery that a large number of chloroplast proteins exhibit dynamic accumulation pattern under drought stresses. The translocation experiments highlight the potential importance of chloroplast proteins in retrograde signaling pathways and provide a set of candidate proteins.

## Figures and Tables

**Figure 1 cells-09-00259-f001:**
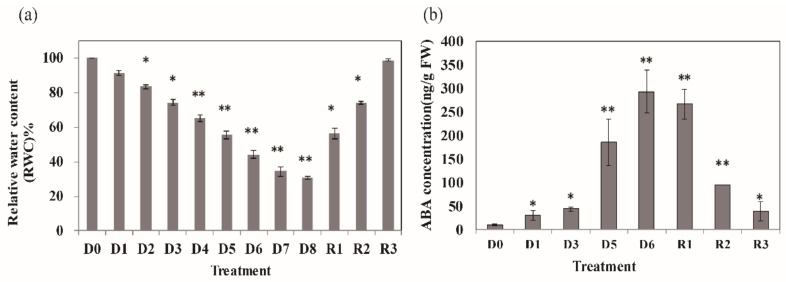
Effects of drought and re-watering treatment on *N. benthamiana* (**a**) Relative water content (RWC) of *N. benthamiana* under drought and re-watering treatment. (**b**) Phytohormones (ABA) content of *N. benthamiana* leaves under drought and re-watering treatment. Data are represented as means ± SD, n = 3; asterisks indicate the values of treatment different from control (WT) ** *p* < 0.01, * *p* < 0.05 (*t-test*).

**Figure 2 cells-09-00259-f002:**
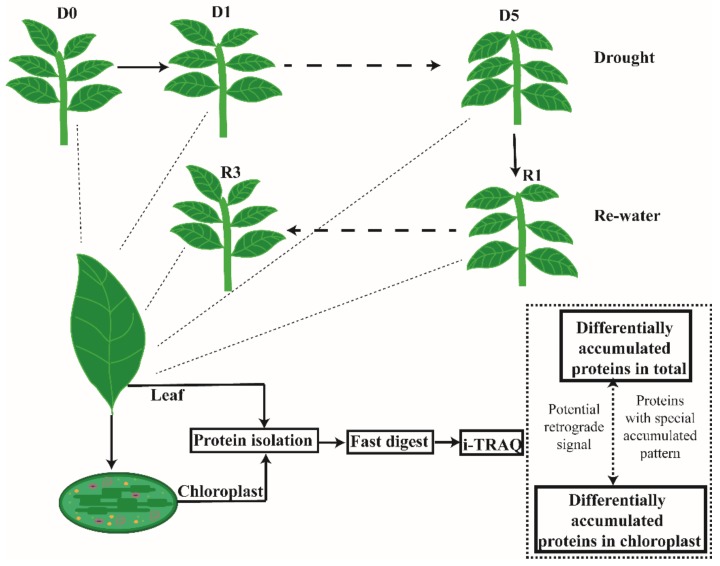
Treatment and analysis work flow.Drought and re-watering treatment were imposed on *N. benthamiana* seedlings; D0 (no-drought), D1 (first day of drought), D5 (fifth day of drought), R1 (8 h of re-watering) and R3 (48 h of re-watering) samples were used for protein isolation, fast-digest, iTRAQ and further analysis.

**Figure 3 cells-09-00259-f003:**
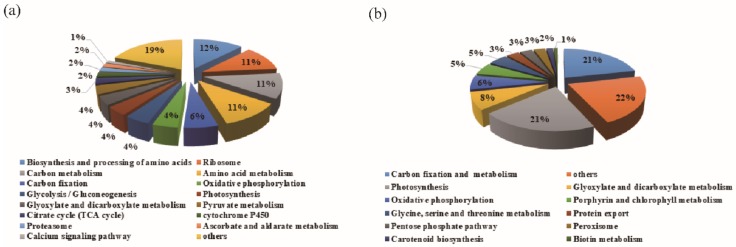
KEGG enrichment of differentially abundant proteins in leaves and chloroplasts. The KEGG pathways significantly enriched among differentially abundant (**a**) total proteins (TPs) and (**b**) chloroplast proteins (CPs) with *p*-value < 0.05.

**Figure 4 cells-09-00259-f004:**
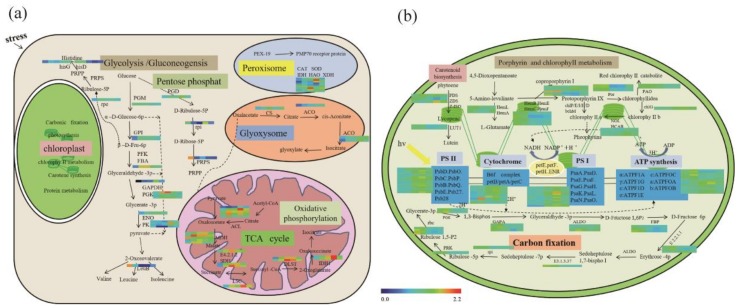
Differentially abundant proteins and associated pathways under drought stress. (**a**) The diverse pathways in leaf showing significant response to drought stress: proteins in glycolysis/gluconeogenesis, pentose phosphate, histidine synthesis, oxidative phosphorylation, and the TCA cycle were significantly induced by drought stress. Glucose is the precursor of pentose phosphate and glycolysis/gluconeogenesis metabolism. D-Fru-6p produced by glycolysis can serve as the main precursor of histidine synthesis. Abbreviations for enzymes: phosphoglucomutase (PGM); glucose-6-phosphate isomerase (GPI); fructose-1,6-bisphosphatase I (FBP); 6-phosphofructokinase (PFK); glyceraldehyde 3-phosphate dehydrogenase (GAPDH); phosphoglycerate kinase (PGK); enolase (ENO) and pyruvate kinase (PK); 6-phosphogluconate dehydrogenase (PGD); 6-phosphogluconate dehydrogenase (rpi); ribose-phosphate pyrophosphokinase (PRPS); fumarate hydratase, class I (E4.2.1.2); succinyl-CoA synthetase alpha subunit (LSC); succinate dehydrogenase (ubiquinone) flavoprotein subunit (SDH); ATP citrate (pro-S)-lyase (ACL); malate dehydrogenase (MDH); dihydrolipoamide succinyltransferase (DLST); icd; isocitrate dehydrogenase (IDH1); ATP phosphoribosyltransferase (hisG, hisD); ribulose-phosphate 3-epimerase (rpe); ribose-phosphate pyrophosphokinase(PRPS); 3-isopropylmalate dehydrogenase (leuB); catalase (CAT); superoxide dismutase, Fe-Mn family (SOD); isocitrate dehydrogenase (IDH); (S)-2-hydroxy-acid oxidase (HAO); xanthine dehydrogenase/oxidase (XDH); citrate synthase (CS); aconitate hydratase (ACO). (**b**) The proteins and significantly induced pathways in chloroplasts by drought stress. Most enzymes of carbon fixation, photosynthesis, porphyrin, and chlorophyll metabolism and carotenoid biosynthesis were significantly induced by drought stress. Abbreviations for enzymes: carotene epsilon-monooxygenase (LUT1); uroporphyrinogen decarboxylase (Hem); magnesium chelatase subunit (chl); magnesium-protoporphyrin O-methyltransferase (bchm); protochlorophyllide reductase (por); chlorophyll(ide) b reductase (NOL); photosystem I subunit (Psa); ferredoxin--NADP+ reductase (pet); photosystem I subunit IV (psa); F-type H+-transporting ATPase subunit (ATPF); phosphoglycerate kinase (PGK); glyceraldehyde-3-phosphate dehydrogenase (NADP+) (GAPA); fructose-bisphosphate aldolase, class I (ALDO); fructose-1,6-bisphosphatase I (FBP); sedoheptulose-bisphosphatase (E3.1.3.31); large subunit ribosomal protein (rpl); 5′-AMP-activated protein kinase, catalytic alpha subunit (PRK); ribulose-bisphosphate carboxylase small chain (rbc). Broken lines indicate possible, but not confirmed, routes. Full lines represent the routes proved in previous studies. Heatmaps of the proteins accumulation related to these pathways are shown.

**Figure 5 cells-09-00259-f005:**
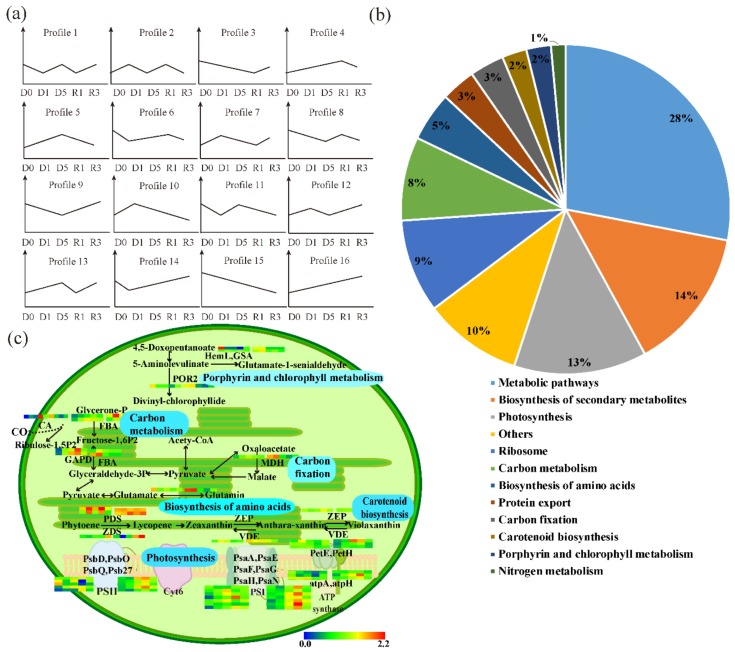
Differential accumulation patterns of TPs and CPs. (**a**) The accumulation patterns of differentially abundant proteins in TPs and CPs. (**b**) The KEGG enrichment analysis of PSAPs (proteins that decreased in chloroplasts but increased in leaves). (**c**) Schematic diagram of the pathways identified in (**b**). The signals produced in the chloroplast appear to induce drought responses in terms of metabolism, including photosynthesis, carbon metabolism, carbon fixation, amino biosynthesis, carotenoid biosynthesis and porphyrin and chlorophyll metabolism. Full lines represent the routes demonstrated in previous studies. Heatmaps for the protein accumulation involved in these pathways are shown, with TPs on the left and CPs on the right.

**Figure 6 cells-09-00259-f006:**
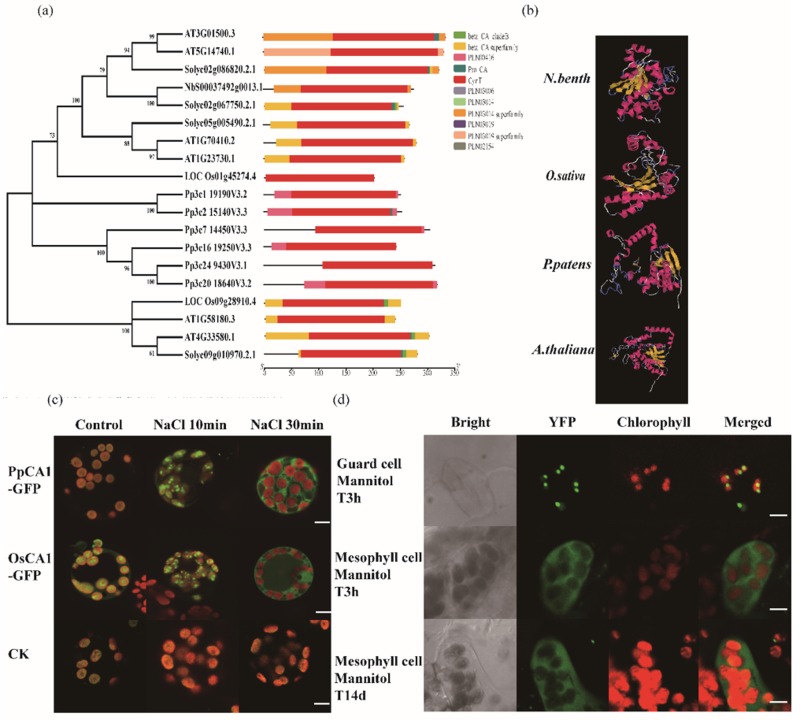
The localization of homologous of *N. benthamiana* CA1 (NbS00037492g0013.1). (**a**) Phylogenetic relationships and motif compositions of NbS00037492g0013.1 and its homologs in A. thaliana, *S. lycopersicum*, *P. patens.* and *O. sativa*. Multiple sequence alignment of NbS00037492g0013.1 and its homologs in *A. thaliana, S. lycopersicum, P. patens* and *O. sativa* using MEGA 6.06 by the NJ method with 1,000 bootstrap replicates (left panel). A schematic representation of conserved motifs were presented in NbS00037492g0013.1 and its homologs in *A. thaliana, S. lycopersicum, P. patens* and *O. sativa*. Motifs were identified by MEME software using complete amino acid sequences of. Different motifs are represented by different colored boxes. Details of the individual motifs are in the right (right panel). (**b**) Protein structures of NbS00037492g0013.1 and its homologs in *P. patens, O. sativa,* and *A. thaliana.* Protein sequence analysis using I-TASSER software showed that the conserved crystalized protein structures (1ddzA, 1ekjG, 401KA, 3ucnA, 5swcA, 5cxkA, 4rxyA, and 2w3qA) and ligand-binding sites (ACT-2w3Na and BCT-5cxkH). (**c**) Localization of NbS00037492g0013.1 homologs in *P. patens* and *O. sativa* under stresses. CK was a RB-GFP (fusing Rubisco transit peptide with GFP), which was chloroplast-located. The proteins were found in the chloroplasts under normal conditions (control: water). After stress treatment (50 mM NaCl) for 10 min, the signal concentrated on some points of the chloroplast, and aggregated in the vicinity of the chloroplast membrane; after more than 30 min stress treatment, some of the signal transported out of the chloroplast, and then translocated to the cell cytosol. The scale bar = 20μm. (**d**) Localization of NbS00037492g0013.1 homologs in A.thaliana. After the detached leaves of CA1 overexpression lines (pGC1: CA-II-YFP) were treated (400 mM Mannitol) for 3 h, the signals concentrated on some points of the chloroplast (top row), transported out of the chloroplast, and then translocated to the cell cytosol (middle row); after more than 14 days treatment to the overexpression lines, the signals obviously transported to the cell cytosol. The scale bar = 20 μm.

**Table 1 cells-09-00259-t001:** The homolog proteins of *N. benthamiana* CA1 (NbS00037492g0013.1) in *P. patens*, *O. sativa*, *A. thaliana* and *S. lycopersicum*.

Protein ID	Species	E-Value	Identity
Pp3c1_19190	*Physcomitrella patens*	1 × 10^80^	0.49
Pp3c16_19250	*Physcomitrella patens*	2 × 10^72^	0.5
Pp3c2_15140	*Physcomitrella patens*	1 × 10^70^	0.41
Pp3c24_9430	*Physcomitrella patens*	13× 10^64^	0.46
Pp3c20_18640	*Physcomitrella patens*	5× 10^64^	0.45
Pp3c7_14450	*Physcomitrella patens*	3 × 10^61^	0.42
LOC_Os01g45274.4	*Oryza sativa*	3 × 10^81^	0.57
LOC_Os09g28910.4	*Oryza sativa*	7× 10^62^	0.45
AT3G01500.3	*Arabidopsis thaliana*	1 × 10^115^	0.64
AT5G14740.1	*Arabidopsis thaliana*	1.7 × 10^113^	0.637
AT1G70410.2	*Arabidopsis thaliana*	13.5× 10^101^	0.563
AT1G23730.1	*Arabidopsis thaliana*	6.3 × 10^97^	0.554
AT1G58180.3	*Arabidopsis thaliana*	5.3 × 10 ^43^	0.367
AT4G33580.1	*Arabidopsis thaliana*	4.8 × 10^54^	0.403
Solyc02g086820.2.1	*Solanum lycopersicum*	2.8 × 10^130^	0.708
Solyc02g067750.2.1	*Solanum lycopersicum*	3.2× 10^105^	0.81
Solyc05g005490.2.1	*Solanum lycopersicum*	1.1 × 10^112^	0.625
Solyc09g010970.2.1	*Solanum lycopersicum*	1.9 × 10^58^	0.497

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
