# Peer review of "Translocation of Drought-Responsive Proteins from the Chloroplasts"

_cells, 2020, doi:10.3390/cells9010259_

Round 1
Reviewer 1 Report
Li et al. reported an interesting study regarding the translocation of drought-responsive proteins from the chloroplasts
The manuscript is well written and easy to read
Line 43. The authors should add a reference.
Line 57. The authors should add a reference.
Line 100. The authors should report who provide the seedlings, the dimension of the pots and the soil used. In addition, the fertilizers and plant protection products used in the cultivations should be added as well.
Line 104. How many pots?
Line 107. All the abbreviations should be defined
Line 113. The authors should briefly report the used and cited method
The figure captions should be stand alone. I suggest to add all the abbreviations showed in the corresponding figure.
I suggest to consider the addition of conclusions section
Author Response
The letter to editor and the point-by-point response to the reviewer’s comments are as follows:
Dear Editor,
On behalf of my co-authors I want to thank you for handling our manuscript. We have carefully read the reviewers’ comments, and modified some figures of the article. Revision has been made based on these comments. The point-by-point response to the reviewer’s comments are outlined in the following.
Reviewer 1
Line 43. The authors should add a reference.
Response: we have added several new references according to reviewer’s suggestions.
Line 57. The authors should add a reference.
Response: we have added reference according to reviewer’s suggestions.
Line 100. The authors should report who provide the seedlings, the dimension of the pots and the soil used. In addition, the fertilizers and plant protection products used in the cultivations should be added as well.
Response: we have revised the sentences to Nicotiana benthamiana seedlings (a gift from Dr. Yongjun Lin in Huazhong Agricultural University) were grown in potting soil with 3L plastic pots. We didn’t use any plant protection products during the culture.
Line 104. How many pots?
Response: we tested at least 5 pots at every treatment point, the values of relative water content were calculated with the mean ± SD (n = 5-7).
Line 107. All the abbreviations should be defined.
Response: we have added the description of “RWC” as follows: Relative water content (RWC) was calculated using the formula: RWC (%) = ((FW-DW)/(TW-DW)) × 100,
Line 113. The authors should briefly report the used and cited method
Response: we have added the description to “The total chlorophyll was extracted from the treatment tobacco leaves with extracted buffer (50% acetone and 50% ethanol) and measured with spectrophotometer (Yiheng, China) [30]”.
The figure captions should be stand alone. I suggest to add all the abbreviations showed in the corresponding figure.
Response: we have separated all the figure captions.
I suggest to consider the addition of conclusions section
Response: we have added it.

Reviewer 2 Report
A very well written manuscript but some information is lacking and it needs to be added prior to publication. This is outlined in the comments below.
Abstract.
L25: should read ‘retrograde signal transduction’ not ‘signals’.
L28: Differentially accumulated is not the correct term. iTRAQ measures the difference in abundance of peptides between samples to infer a difference is abundance of a protein (actually an ORF product).
L29: should read ‘one particular category of proteins, that includes carbonic anhydrase 1 (CA1), exhibited a great decline in chloroplasts’.
L33: ‘consolidated’ is not the correct term. The sentence needs to be reworded to be point out that the subcellular localisation experiments were separate experiments that demonstrated that CA1 abundance changed due to the treatment. Also, replace CA1 proteins with CA1 homologs to maintain clarity.
L34: should read ‘accumulated in the cytosol’ rather than cytosols.
Introduction.
L48: Need to reword sentence to read ‘through communication pathways whose mechanism is unclear’.
L50: should read ‘it poses the possibility’.
L72: While it seems like I am being picky and semantic, iTRAQ is NOT quantifying the abundance of proteins. It is quantifying the abundance of peptides that are then used to INFER the abundance of the products of an Open Reading Frame (ORF). This has implications for answering the question posed on L64, the answer to which could be that proteoform differences are present and these are not able to be measured by iTRAQ.
L73: change ‘protein expression’ to ‘protein abundance’.
L74: ‘The proteomics studies in maize (Zea mays), cotton (Gossypium sp.), poplar (Populus 74 cathayana), and soybean (Glycine max) have established that iTRAQ is effective and reliable’. Effective and reliable for what? Sentence is incomplete.
L82: ‘orchestrating’ is the wrong word. Should be ‘understanding’ or something similar.
L85: Change to ‘differential abundance’.
L91: delete the word ‘candidate’.
L92: Can you clarify whether the proteins were induced by drought or whether they had moved from the chloroplast to the cytosol? Was the amount of proteins in the entire leaf the same when comparing control to treatment, but less present in the chloroplast of the treatment, indicating movement from chloroplast to cytosol?
How does subjecting the protoplasts to osmotic stress simulate the changes that occur in drought treatment?
Methods.
State the amount and volume of water used for rewatering.
Please provide the details of the LC/MS/MS experiment including solvents, gradient conditions, column type and dimensions, ionisation voltage, MS1 and MS2 acquisition parameters. In addition, the parameters for the Mascot search must be stated as well as how the quantification of peptide abundances from the iTRAQ reporter ions was performed and any statistical analysis was performed.
L159: ‘The intensity values of ion peaks were drawn’. Are these the intensity of the iTRAQ reporter ions? Also, ‘drawn’ is not the right word. Should be ‘extracted’.
L172: phylogenetic tree? From protein sequence data?
Results.
L206: ‘Accumulation’ is not the correct word because some proteins will have a reduction in amount or abundance and thus they are not accumulating.
L210: should read ‘Purity assessment of chloroplast proteins and total proteins was performed through Western blot with antibodies specific for Rubisco and AOX’.
L212: change ‘accumulating’.
L213: Of the proteins identified, how many were single peptide identifications (one-hit wonders)?
L218: Please elaborate on the statement that many of the differentially abundant proteins were related to ABA. How are they related? In the synthesis pathway? Something else?
Can the authors reword the description of Table S1. The labels on the columns in the table don’t match with that in the text and it is confusing.
L220: should read ‘reported to be involved’. And change ‘accumulation’ unless those proteins are increasing in their abundance.
L221: ‘We found that 39 ROS-associated proteins, 47 HSPs and 29 transcription factors also participated in the drought stress responses’. This is incorrect. You have found that the abundance is altered. Participation is not the correct word.
Paragraph from L228: Please change all instances of ‘accumulated’.
L246: ‘However, the quantity and fold change were different, with CPs showing larger changes than TPs (Table S1).’ This is almost impossible to determine from the table as presented. Please provide some notable examples or reformat the table so that this observation can be seen.
Figure 4: The heat maps need a key to show what is increasing and what is decreasing.
L284: The proteins are not induced. They are increasing in abundance which IMPLIES that gene expression is induced (which has not been measured).
L289: ‘The related metabolism and proteins were likely induced via signals produced by glycolysis during drought stress.’ While this is likely true, there is no evidence presented to support this statement. You should provide references that support this statement.
L305: ‘This was not surprising because these proteins act as the upstream precursors for synthesizing stress regulators like ABA, JA, and others.’ This statement requires references and some elaboration. Proteins are not used as precursors for the production of ABA, so the statement is incorrectly worded.
Figure 5: Need a key for the heat maps.
L362: The sentence implies that the complete gene, and thus ORF, sequence for CA1 is not known. Thus it has been assumed that the ORF sequence from moss and rice is exactly the same but this may not be the case and sequence differences could be causing proteoform differences. This needs clarifying in the text.
Table 1: Does ‘identity’ refer to sequence homology? Please clarify.
Discussion.
L421: Can the authors comment on the large discrepancy between this study and reference 32.
L450: ‘This study is in accordance with previous reports that leaf senescence and stress 450 responses share common pathways involving cell death.’ Provide references.
This experiment needs to be repeated in moss, rice or Arabidopsis to confirm the observations in a better characterised system. This should be noted as a future direction.
Author Response
The letter to editor and the point-by-point response to the reviewer’s comments are as follows:
Dear Editor,
On behalf of my co-authors I want to thank you for handling our manuscript. We have carefully read the reviewers’ comments, and modified some figures of the article. Revision has been made based on these comments. The point-by-point response to the reviewer’s comments are outlined in the following.
Reviewer 2
Abstract. L25: should read ‘retrograde signal transduction’ not ‘signals’.
Response: we have revised it.
Abstract. L28: Differentially accumulated is not the correct term. iTRAQ measures the difference in abundance of peptides between samples to infer a difference is abundance of a protein (actually an ORF product).
Response: we have revised all the Differentially accumulated proteins to Differentially abundant proteins.
Abstract. L29: should read ‘one particular category of proteins, that includes carbonic anhydrase 1 (CA1), exhibited a great decline in chloroplasts’.
Response: we have revised it.
Abstract. L33: ‘consolidated’ is not the correct term. The sentence needs to be reworded to be point out that the subcellular localisation experiments were separate experiments that demonstrated that CA1 abundance changed due to the treatment. Also, replace CA1 proteins with CA1 homologs to maintain clarity.
Response: we have reworded the sentence to “The subcellular localizations of CA1 proteins from moss (Physcomitrella patens), Arabidopsis thaliana and rice (Oryza sativa) in P. patens protoplasts consistently showed that CA1 proteins gradually diminished within chloroplasts but increasingly accumulated in the cytosol under osmotic stress treatment, suggesting that they could be translocated from chloroplasts to the cytosol and act as a signal messenger from the chloroplast.”
Abstract. L34: should read ‘accumulated in the cytosol’ rather than cytosols.
Response: we have reworded the sentence above.
Introduction.L48: Need to reword sentence to read ‘through communication pathways whose mechanism is unclear’.
Response: we have revised it.
Introduction.L50: should read ‘it poses the possibility’.
Response: we have revised it.
Introduction.L72: While it seems like I am being picky and semantic, iTRAQ is NOT quantifying the abundance of proteins. It is quantifying the abundance of peptides that are then used to INFER the abundance of the products of an Open Reading Frame (ORF). This has implications for answering the question posed on L64, the answer to which could be that proteoform differences are present and these are not able to be measured by iTRAQ.
Response: we have revised the sentence to Isobaric tags for relative and absolute quantitation (iTRAQ) is an isobaric labeling method coupled with tandem mass spectrometry for proteome analysis.
Introduction.L73: change ‘protein expression’ to ‘protein abundance’.
Response: we have revised it.
Introduction. L74: ‘The proteomics studies in maize (Zea mays), cotton (Gossypium sp.), poplar (Populus 74 cathayana), and soybean (Glycine max) have established that iTRAQ is effective and reliable’. Effective and reliable for what? Sentence is incomplete.
Response: we have completed it with “method for proteome analysis under stress”.
Introduction. L82: ‘orchestrating’ is the wrong word. Should be ‘understanding’ or something similar.
Response: we have advised it to elucidating.
Introduction. L85: Change to ‘differential abundance’.
Response: we have revised it.
Introduction. L91: delete the word ‘candidate’.
Response: we have deleted it.
Introduction. L92: Can you clarify whether the proteins were induced by drought or whether they had moved from the chloroplast to the cytosol? Was the amount of proteins in the entire leaf the same when comparing control to treatment, but less present in the chloroplast of the treatment, indicating movement from chloroplast to cytosol?
Response: The abundance of proteins increased on total and chloroplast under drought stress when comparing control to treatment, then 231 proteins decreased in chloroplasts during the end of drought treatment or the beginning of the re-watering treatment, but increased in leaf cells at the same time (Figure 5b, Table S5). The confocal microscopic imaging of CA1 is to validate the responses of these identified proteins to drought stress, which move from chloroplast to cytosol under stress treatment.
Introduction. How does subjecting the protoplasts to osmotic stress simulate the changes that occur in drought treatment?
Response: we choose NaCl or Mannitol to mimic the drought stress on plants, because of the following reasons:
1) the treatment concentration was generally used in Arabidopsis and other plants (Kim et al., 2007; Kugler et al., 2009; Yan et al., 2014; Ha et al., 2014).
2) we used Mannitol and NaCl to mimic drought stress, because these three abiotic stresses often induce a highly overlapped genome wide changes in the large number of transcriptome data (Chen et al., 2019; Yin et al., 2019).
3) according to the previous reports, drought related critical genes such as Arabidopsis AtbZIP1, poplar SCL7, cotton WRKY17, tobacco LTP4 or rice PP2C are also responsive to salt or osmotic stresses (Ma et al., 2010; Sun et al., 2012; Ha et al., 2014; Yan et al., 2014; Singh et al., 2015; Xu et al., 2018 ).
Methods.State the amount and volume of water used for rewatering.
Please provide the details of the LC/MS/MS experiment including solvents, gradient conditions, column type and dimensions, ionisation voltage, MS1 and MS2 acquisition parameters. In addition, the parameters for the Mascot search must be stated as well as how the quantification of peptide abundances from the iTRAQ reporter ions was performed and any statistical analysis was performed.
Response: the fully absorbing water method was used for rewatering, during which we added water until it drains from the bottom of pots. The ITRAQ labeling and LC-MS/MS assay were completed by the company of Shanghai Applied Protein Technology with the details on www.aptbiotech.com.
Methods.L159: ‘The intensity values of ion peaks were drawn’. Are these the intensity of the iTRAQ reporter ions? Also, ‘drawn’ is not the right word. Should be ‘extracted’.
Response: we have revised it.
Methods.L172: phylogenetic tree? From protein sequence data?
Response: Yes, the phylogenetic tree was generated using protein sequences and MEGA 6.06 with the Neighbor-Joining (NJ) method.
Results.L206: ‘Accumulation’ is not the correct word because some proteins will have a reduction in amount or abundance and thus they are not accumulating.
Response: we have revised all the Differentially accumulated proteins to Differentially abundant proteins.
Results.L210: should read ‘Purity assessment of chloroplast proteins and total proteins was performed through Western blot with antibodies specific for Rubisco and AOX’.
Response: we have revised it.
Results.L212: change ‘accumulating’.
Response: we have revised all the Differentially accumulated proteins to Differentially abundant proteins.
Results.L213: Of the proteins identified, how many were single peptide identifications (one-hit wonders)?
Response: during the identified 5,230 TPs and 1,522 CPs, 2462 TPs and 751CPs were with one Unique Peptides.
Results.218: Please elaborate on the statement that many of the differentially abundant proteins were related to ABA. How are they related? In the synthesis pathway? Something else?
Response: From the i-TRAQ sequencing results, many proteins related to ABA were reflected by the drought and re-water treatment. In table 1R, we selected some proteins related to ABA metabolism and signaling, and found that they were significantly reflected as follows:
ABA metabolism:
NCED: a rate-limiting biosynthetic enzyme for ABA production, it was accumulated during drought stress(W-S2) and decreased during re-water(R1) treatment (TPs); decreased during first drought stress (W-S1), accumulated during last stress(S1-R0), decreased in re-water(R1) (CPs).
ABA signaling:
RAB: decreased during the first drought stress(S0), increased then(TPs); which increased in drought stress(CPs).
CYP: increased during the drought stress and decreased at the end(re-water) (TPs and CPs).
Table 1R Proteins related to ABA
|
Accession |
LD0 |
LS1 |
LS2 |
LR1 |
LR2 |
p_value |
key_word |
|
NbS00008382g0001.1 |
1.251387623 |
0.842176 |
1.022734 |
1.260306 |
1.236533 |
0.247091 |
RABD |
|
NbS00004422g0017.1 |
1.099715894 |
1.133557 |
1.25688 |
1.158315 |
1.117486 |
0.0154339 |
SAPK |
|
NbS00000156g0020.1 |
0.892302011 |
0.937516 |
1.05886 |
1.159127 |
1.060278 |
0.0240053 |
NCED |
|
NbS00055510g0007.1 |
0.988656916 |
0.866465 |
1.02059 |
1.059432 |
0.836515 |
0.0979443 |
NCED |
|
NbS00004558g0003.1 |
0.883317584 |
1.02252 |
1.120019 |
1.191421 |
0.975911 |
0.100839 |
CYP |
|
NbS00000477g0007.1 |
0.663999153 |
0.818389 |
0.801797 |
0.856844 |
0.898201 |
0.431755 |
CYP |
|
NbS00040365g0001.1 |
0.618520497 |
0.669611 |
0.683543 |
0.976807 |
0.758084 |
0.838845 |
CYP |
|
NbS00035145g0007.1 |
0.85936534 |
0.939289 |
1.048127 |
0.890887 |
0.886876 |
0.164945 |
CTSB |
|
NbS00022651g0005.1 |
0.912022643 |
1.119972 |
0.991946 |
1.006325 |
0.926632 |
0.0289134 |
SUE |
|
NbS00032668g0001.1 |
0.805932258 |
0.823347 |
0.835672 |
0.788985 |
0.665488 |
0.918427 |
DUF |
|
NbS00008667g0012.1 |
1.076847681 |
0.923181 |
0.992895 |
0.797329 |
0.79074 |
0.484533 |
SL2 |
|
NbS00056196g0007.1 |
0.800855704 |
1.108575 |
1.047323 |
1.051142 |
1.02713 |
0.0024005 |
EF- |
|
NbS00004333g0013.1 |
0.878083881 |
0.91972 |
0.91553 |
0.891004 |
1.041187 |
0.0651971 |
SnRK |
|
NbS00023323g0005.1 |
1.055221428 |
0.944643 |
0.929722 |
0.956274 |
0.877471 |
0.945528 |
CYP |
|
Accession |
CD0 |
CS1 |
CS2 |
CR1 |
CR2 |
p_value |
key_word |
|
NbS00015470g0013.1 |
0.804149681 |
0.9334 |
1.176917 |
1.3518 |
1.122675 |
6.762E-06 |
RAB |
|
NbS00027807g0002.1 |
1.163120861 |
1.090468 |
0.941848 |
0.846609 |
0.922483 |
2.04E-05 |
RABE |
|
NbS00019210g0010.1 |
0.705450542 |
0.852098 |
0.998025 |
1.17433 |
1.005823 |
0.126188 |
RAB |
|
NbC23171932g0001.1 |
1.037879689 |
1.105497 |
1.057553 |
0.829462 |
0.931248 |
0.588292 |
RAB |
|
NbS00055510g0007.1 |
0.71033223 |
0.703265 |
1.121403 |
1.145746 |
0.97111 |
0.0051316 |
NCED |
|
NbS00022310g0012.1 |
1.05507537 |
0.969783 |
0.984837 |
1.252812 |
0.887437 |
0.0883825 |
SEP |
|
NbS00004558g0003.1 |
0.739785373 |
0.824262 |
1.394333 |
1.130076 |
1.486369 |
2.836E-06 |
CYP |
|
NbS00037672g0006.1 |
0.775860069 |
0.837083 |
1.368408 |
1.373258 |
1.360018 |
0.0011377 |
CYP |
|
NbS00060756g0002.1 |
0.830407999 |
0.875336 |
1.146071 |
1.126518 |
1.292591 |
0.005806 |
CYP |
|
NbS00012214g0002.1 |
0.906011136 |
0.894969 |
1.063533 |
1.226723 |
1.180806 |
0.0575196 |
CYP |
|
NbS00005095g0002.1 |
1.230879248 |
0.918653 |
0.690698 |
0.56517 |
0.741035 |
3.814E-05 |
AHRD |
|
NbS00001586g0015.1 |
0.993834046 |
1.105531 |
0.916403 |
1.001835 |
1.020017 |
0.0181925 |
IMS1 |
|
NbS00026065g0006.1 |
1.240468573 |
0.99323 |
0.963986 |
0.786347 |
0.917986 |
0.029393 |
ATIMD2 |
Results.Can the authors reword the description of Table S1. The labels on the columns in the table don’t match with that in the text and it is confusing.
Response: we have completed the description of Table S1 as follows: Table S1. All differentially abundant proteins (DAPs) in leaves (LD0, LS1, LS2, LR1, LR2) and chloroplasts (CD0, CS1, CS2, CR1, CR2) under drought stress (FDR< 0.05 and log2|fold change| >1).
Results.L220: should read ‘reported to be involved’. And change ‘accumulation’ unless those proteins are increasing in their abundance.
Response: we have revised and checked it.
Results.L221: ‘We found that 39 ROS-associated proteins, 47 HSPs and 29 transcription factors also participated in the drought stress responses’. This is incorrect. You have found that the abundance is altered. Participation is not the correct word.
Response: we have revised it to “We found that 39 ROS-associated proteins, 47 HSPs and 29 transcription factors also affected by drought stress”.
Results.Paragraph from L228: Please change all instances of ‘accumulated’.
Response: we have revised it
Results.L246: ‘However, the quantity and fold change were different, with CPs showing larger changes than TPs (Table S1).’ This is almost impossible to determine from the table as presented. Please provide some notable examples or reformat the table so that this observation can be seen.
Response: we have revised it to “However, the quantity and fold change were different, with most CPs showing larger changes than TPs, such as OHP2 (NbC25835994g0001.1), PSAF (NbS00000058g0018.1), and so on (Table S1)”.
Results.Figure 4: The heat maps need a key to show what is increasing and what is decreasing.
Response: we have added it.
Results.L284: The proteins are not induced. They are increasing in abundance which IMPLIES that gene expression is induced (which has not been measured).
Response: we have revised it to “As Figure 4a shows, proteins related to glycolysis/gluconeogenesis metabolism were significantly accumulated under drought stress.”
Results.L289: ‘The related metabolism and proteins were likely induced via signals produced by glycolysis during drought stress.’ While this is likely true, there is no evidence presented to support this statement. You should provide references that support this statement.
Response: we have deleted the sentence.
Results.L305: ‘This was not surprising because these proteins act as the upstream precursors for synthesizing stress regulators like ABA, JA, and others.’ This statement requires references and some elaboration. Proteins are not used as precursors for the production of ABA, so the statement is incorrectly worded.
Response: we have revised it to “This was not surprising because the products of porphyrin and chlorophyll metabolism act as the upstream precursors for synthesizing stress regulators like ABA, JA, and others ” (https://www.kegg.jp/) .
Results.Figure 5: Need a key for the heat maps.
Response: we have added it.
Results.L362: The sentence implies that the complete gene, and thus ORF, sequence for CA1 is not known. Thus it has been assumed that the ORF sequence from moss and rice is exactly the same but this may not be the case and sequence differences could be causing proteoform differences. This needs clarifying in the text.
Response: we have revised it to “Six and two homologous proteins were found from moss (PpCA1) and rice (OsCA1) using the signature peptide of NbCA1, respectively”.
Results.Table 1: Does ‘identity’ refer to sequence homology? Please clarify.
Response: yes, the ‘identity’ refer to sequence homology between proteins from different specials and the signature peptide of NbCA1.
Discussion.L421: Can the authors comment on the large discrepancy between this study and reference 32.
Response: we want to say that we have similar KEGG pathway analyses with the reference.
Discussion.This experiment needs to be repeated in moss, rice or Arabidopsis to confirm the observations in a better characterised system. This should be noted as a future direction.
Response: thank you for your suggestion, we will put it as the future work.
References:
Chen, N., Feng, J., Song, B., Tang, S., He, J., & Zhou, Y., et al. (2019). De novo transcriptome sequencing and identification of genes related to salt and peg stress in tetraena mongolica maxim. Trees, 1-18.
Ha, C.V., Leyva-Gonzalez, M.A., Osakabe, Y., Tran, U.T., Nishiyama, R., & Watanabe, Y., et al. (2014). Positive regulatory role of strigolactone in plant responses to drought and salt stress. Proceedings of the National Academy of Sciences, 111(2), 851-856.
Kim, B.G., Waadt, R., Cheong, Y.H., Pandey, G.K., Dominguez-Solis, J.R., & Stefanie Schültke, et al. (2007). The calcium sensor cbl10 mediates salt tolerance by regulating ion homeostasis in Arabidopsis. Plant Journal, 52(3), 473-484.
Kugler, A., KÖHler, B., Palme, K., Wolff, P., & Dietrich, P. (2009). Salt-dependent regulation of a cng channel subfamily in Arabidopsis. BMC Plant Biology, 9(1), 140-0.
Ma, H.S., Liang, D., Shuai, P., Xia, X.L., & Yin, W.L. (2010). The salt- and drought-inducible poplar gras protein scl7 confers salt and drought tolerance in Arabidopsis thaliana. Journal of Experimental Botany, 61(14), 4011-4019.
Singh, A., Jha, S.K., Bagri, J., Pandey, G.K. (2015). ABA Inducible Rice Protein Phosphatase 2C Confers ABA Insensitivity and Abiotic Stress Tolerance in Arabidopsis. PLoS ONE 10(4): e0125168.
Sun, X., Li, Y., Cai, H., Bai, X., Ji, W., & Ding, X., et al. (2012). The Arabidopsis at bzip1 transcription factor is a positive regulator of plant tolerance to salt, osmotic and drought stresses. Journal of Plant Research, 125(3), 429-438.
Xu, Y., Zheng, X., Song, Y., Zhu, L., Yu, Z., & Gan, L., et al. (2018). Ntltp4, a lipid transfer protein that enhances salt and drought stresses tolerance in Nicotiana tabacum. Scientific Reports, 8(1), 8873.
Yan, H., Jia, H., Chen, X., Hao, L., An, H., & Guo, X. (2014). The cotton wrky transcription factor ghwrky17 functions in drought and salt stress in transgenic Nicotiana benthamiana through ABA signaling and the modulation of reactive oxygen species production. Plant and Cell Physiology, 55(12), 2060-2076.
Yin, H., Li, M., Li, D., Khan, S. A., Hepworth, S. R., & Wang, S. M. (2019). Transcriptome analysis reveals regulatory framework for salt and osmotic tolerance in a succulent xerophyte. BMC Plant Biology, 19(1).

Reviewer 3 Report
The paper presents the results of identification of drought-dependent proteins in leafs and chloroplasts of Nicotiana benthamiana. These data are of high importance since revealing of molecular mechanisms underlying drought-tolerance could give promising perspectives for future agriculture.
The title of the paper needs to be more detailed. Probably, 'Quantitative proteomic analysis revealed drought-dependent translocation of functionally-different proteins from the chloroplasts' is better (though other variants are also possible).
The Introduction is well-written and of sufficient quality.
The Results are clearly presented but several additions should be made prior publication.
First, for each experiment a number of repeats must be indicated. It can either be incorporated in Materials and Methods or be shown in each subsection of Results.
Also, Figures 3,4,5 are difficult for analysis since their parts have little sizes and contain a number of details (though they are beautiful). I suppose to relocate parts A and B of Figure 3 one above the other and stretch across the width of the page. This will also be useful for Figure 4. The Figure Five should be separated onto Figures 5(a-b) and 6(c) to increase a visual quality.
Please, change 'Gen' to 'Gene' (line 163).
Conclusion paragraph (lines 488-492) should be expanded. Please, include most important findings of the paper (brief description of functional groups of translocated proteins, findings with CA1, etc).
Author Response
The letter to editor and the point-by-point response to the reviewer’s comments are as follows:
Dear Editor,
On behalf of my co-authors I want to thank you for handling our manuscript. We have carefully read the reviewers’ comments, and modified some figures of the article. Revision has been made based on these comments. The point-by-point response to the reviewer’s comments are outlined in the following.
Reviewer 3
The title of the paper needs to be more detailed. Probably, 'Quantitative proteomic analysis revealed drought-dependent translocation of functionally-different proteins from the chloroplasts' is better (though other variants are also possible).
Response: thank you for your suggestion.
The Introduction is well-written and of sufficient quality.
Response: thank you for your encouragement.
The Results are clearly presented but several additions should be made prior publication.
Response: thank you for your encouragement and we have added some details.
First, for each experiment a number of repeats must be indicated. It can either be incorporated in Materials and Methods or be shown in each subsection of Results.
Response: we have checked and added it during the Materials and Methods.
Also, Figures 3,4,5 are difficult for analysis since their parts have little sizes and contain a number of details (though they are beautiful). I suppose to relocate parts A and B of Figure 3 one above the other and stretch across the width of the page. This will also be useful for Figure 4. The Figure Five should be separated onto Figures 5(a-b) and 6(c) to increase a visual quality.
Response: thank you for your suggestion, because of the large amount of information during the figures, it’s difficult to change the pictures during such a short time, we have relocated parts A and B of Figure 3 and 4 as follows:
Please, change 'Gen' to 'Gene' (line 163).
Response: we have revised it and sorry for our missing.
Conclusion paragraph (lines 488-492) should be expanded. Please, include most important findings of the paper (brief description of functional groups of translocated proteins, findings with CA1, etc).
Response: we have added the conclusion section.
